# Clinical Next Generation Sequencing Application in Mesothelioma: Finding a Golden Needle in the Haystack

**DOI:** 10.3390/cancers15245716

**Published:** 2023-12-06

**Authors:** Luigi Cerbone, Sara Orecchia, Pietro Bertino, Sara Delfanti, Antonina Maria de Angelis, Federica Grosso

**Affiliations:** 1Mesothelioma Unit, SS Antonio e and Biagio Hospital, 15121 Alessandria, Italy; luigi.cerbone@ospedale.al.it (L.C.); pietro.bertino@ospedale.al.it (P.B.); sara.delfanti@ospedale.al.it (S.D.); antonina.deangelis@ospedale.al.it (A.M.d.A.); 2Molecular Pathology Unit, SS Antonio e and Biagio Hospital, 15121 Alessandria, Italy; sorecchia@ospedale.al.it

**Keywords:** mesothelioma, target therapy, next generation sequencing

## Abstract

**Simple Summary:**

Mesothelioma is a rare cancer arising from the mesothelium, the epithelial lining of the pleura, peritoneum, pericardium and tunica vaginalis testis. Mesothelioma is a disease with poor prognosis and few targeted treatment options. The genomic landscape of mesothelioma is characterized by the inactivation of oncosuppressor genes, and activating targetable mutations in oncogenes are rare. The goal of this review is to summarize the advancements in the use of clinical next generation sequencing in mesothelioma to guide the use of targeted agents.

**Abstract:**

Mesothelioma comprises a group of rare cancers arising from the mesothelium of the pleura, peritoneum, tunica vaginalis testis and pericardium. Mesothelioma is generally associated with asbestos exposure and has a dismal prognosis, with few therapeutic options. Several next generation sequencing (NGS) experiments have been performed on mesothelioma arising at different sites. These studies highlight a genomic landscape mainly characterized by a high prevalence (>20%) of genomic aberrations leading to functional losses in oncosuppressor genes such as BAP1, CDKN2A, NF2, SETD2 and TP53. Nevertheless, to date, evidence of the effect of targeting these alterations with specific drugs is lacking. Conversely, 1–2% of mesothelioma might harbor activating mutations in oncogenes with specifically approved drugs. The goal of this review is to summarize NGS applications in mesothelioma and to provide insights into target therapy of mesothelioma guided by NGS.

## 1. Introduction

Mesothelioma is a rare cancer arising from the mesothelium, the epithelial lining of the serous membranes. It most often arises in the pleura and less frequently in the peritoneum, pericardium and tunica vaginalis testis [1]. Mesothelioma is correlated with asbestos exposure, and several Western countries have enacted measures against asbestos use to reduce mesothelioma incidences. Nevertheless, the incidence and prevalence of mesothelioma are estimated to rise due to increasing asbestos use in non-Western Countries [2]. Systemic therapy is the backbone of the treatment for most mesothelioma patients, as they are most frequently diagnosed at a late stage [3].

Pleura is the most common site at which mesothelioma arises. Hence, pleural mesothelioma is the only site of mesothelioma for which prospective phase III studies exist in support of systemic treatment application. Thus, most of the evidence of the treatment of mesothelioma arising in other sites is borrowed from the therapeutic schemes used in pleural mesothelioma. Few advances have been made in the treatment of pleural mesothelioma. A combination of platinum and pemetrexed has been the main therapeutic standard since 2003 [4], while a combination of the anti-CTLA-4 monoclonal antibody ipilimumab and the anti-PD1 antibody nivolumab recently showed significantly prolonged OS [5], especially in non-epithelioid histology. In addition, drugs targeting VEGF/VEGFR2, for example, bevacizumab [6] and ramucirumab [7], have shown a certain degree of efficacy in prolonging PFS in pleural mesothelioma and are thus available in several European countries for the treatment of this disease.

The evidence that in some cancers aberrant growth is selectively guided by activating genomic aberrations [8] (i.e., hotspot mutations and fusions) in oncogenes that can be selectively targeted has led to the development of several drugs, including small tyrosine kinase inhibitors, monoclonal antibodies and antibody–drug conjugates. These drugs reshaped the treatment landscape of non-small cell lung cancer [9], melanoma [10] and cholangiocarcinoma [11], to cite a few. To date, no such revolution has occurred in mesothelioma [12].

The goal of this review is to summarize the available evidence on the application of next generation sequencing techniques in guiding the use of targeted agents against mesothelioma.

## 2. The Genomic Landscape of Mesothelioma

### 2.1. Pleural Mesothelioma

#### 2.1.1. Whole Genomic Studies

Given the rarity of the disease, only a few studies have explored the genomic landscape of pleural mesothelioma to date. Comprehensive multi-omics studies of pleural mesothelioma (PM) describe a genomic landscape mostly characterized by a lower number of somatic mutations inducing protein modifications and, conversely, a high level of copy number aberrations leading to functional losses in oncosuppressor genes such as *BAP1*, *NF2*, *CDKN2A*, *SETD2* and *TP53*. The first published study in which whole exome sequencing (WES) was used to analyze 22 tumoral samples from patients [13] affected by PM found recurrent mutations causing protein modifications in *BAP1*, *NF2* and *CUL1*, as well as copy number aberration events leading to loss of function in *CDKN2A*, *NF2* and *BAP1.*

A further study by Bueno et al. [14] investigated the mutational landscape of 216 mesothelioma samples obtained from different patients. Of these samples, 211 were analyzed in whole transcriptomic studies, 103 in targeted exome studies and 99 in whole exome sequencing studies. Using transcriptomic analysis, PM samples were classified into four subgroups: sarcomatoid, epithelioid, biphasic-epithelioid (biphasic-E) and biphasic-sarcomatoid (biphasic-S). These four subgroups had prognostic relevance as the epithelioid cluster showed better overall survival compared with the other groups. Mutational analysis using WES and parallel targeted sequencing identified a low number of protein-coding mutations, as well as a low level of tumor mutational burden (TMB). The most frequently mutated genes in PM were *BAP1*, *NF2*, *TP53*, *SETD2*, *DDX3X*, *ULK2*, *RYR2*, *CFAP45*, *SETDB1* and *DDX51* [14]. Later, a TCGA study led by Hmeljak [15] was performed on 74 PM samples. A four-cluster survival predictor was also developed in this study and showed that epithelial-to-mesenchymal transition (EMT) gene overexpression was associated with a worse prognosis in PM. Genomic sequencing identified *BAP1*, *NF2*, *TP53*, *LATS2* and *SETD2* as significantly mutated genes. In this study, EMT and *VISTA* expression were also identified as valuable targets for tumor treatment as PM was the second most common tumor based on EMT rate and the tumor with the most abundant *VISTA* expression (Summarized in Table 1).

#### 2.1.2. Targeted Next Generation Sequencing (NGS) Studies of Pleural Mesothelioma (PM)

Targeted NGS might be helpful in identifying potentially targetable genomic and transcriptomic alterations. In pleural mesothelioma, a few studies based on targeted NGS have been published so far. A first attempt to identify potentially actionable targets in PM was published by Shukuya et al. in 2014 [16]. In this study, 42 tumor samples from 42 patients affected by PM were analyzed using a targeted amplicon-based panel identifying mutations in cancer-related genes such as *EGFR*, *KRAS*, *BRAF*, *PIK3CA*, *NRAS*, *MEK1*, *AKT1*, *PTEN* and *HER2* and amplifications in *EGFR*, *MET*, *PIK3CA*, *FGFR1* and *FGFR2*. In this study, four patients harbored potentially targetable mutations, three in *PI3KCA* and one in *KRAS*. Lo Iacono et al. then performed a targeted NGS study of 123 PM tumoral samples from patients with at least stage III disease [17] using a custom NGS panel containing 50 known oncogenes (Ion AmpliSeq Cancer Hotspot Panel v. 2, Life Technologies, Grand Island, NY, USA) plus *BAP1* and *NF2*. Despite no targetable alteration being identified in this study [17], patients whose tumors harbored more mutations had significantly worse survival.

A further study conducted by Kato on 42 mesothelioma samples [18] (23 pleural, 11 peritoneal, 2 pericardial, 6 unknown) using a targeted NGS panel able to characterize mutations, copy number alterations (CNAs) and selected fusions in 287 cancer-related genes [19] identified a median of 3 mutations in each tumor included in the study. Furthermore, at least one mutation in each tumor sample was potentially actionable, either with an FDA drug approved in the US for another cancer type or with an investigational drug.

In addition, a study of 43 Brazilian patients with a diagnosis of PM [20], in which a 15-gene (*AKT1*, *GNA11*, *NRAS*, *BRAF*, *GNAQ*, *PDGFRA*, *EGFR*, *KIT*, *PIK3CA*, *ERBB2*, *KRAS*, *RET*, *FOXL2*, *MET* and *TP53*) amplicon-based NGS panel was used to determine the number of potentially actionable mutations, found potentially targetable mutations in *PDGFRA* in 2/43 patients (4.6%). Finally, in the largest NGS-based profiling study to date, Hiltbrunner et al. analyzed NGS data for 1468 patients with mesothelioma, of which 1113 were pleural and 355 were peritoneal mesothelioma cases [21]. The NGS panel used in this study is an updated version of the one previously reported [19] and identifies alterations in 324 cancer-related genes [22]. In this paper, both pleural and peritoneal mesothelioma were confirmed to be characterized by a low number of somatic mutations and a low tumor mutational burden. The most frequent genomic aberrations were copy number alterations (CNAs) leading to functional inactivation mainly in oncosuppressor genes such as *CDKN2A*, *CDKN2B*, *MTAP*, *BAP1* and *NF2*.

Nevertheless, rare (0 to 1%) actionable mutations in genes such as *KRAS*, *EGFR*, *PDGFRA/B*, *ERBB2* and *FGFR3,* as well as >1% mutations in *ALK*, *PTCH1*, *SUFU* and *BRCA2* were found. Matching drugs for all previously reported mutations are commercially available or under experimental testing (Summarized in Table 2).

### 2.2. Peritoneal Mesothelioma (PeM)

Primary peritoneal mesothelioma is the second most common form of mesothelioma. It is a very rare disease that tends to develop at a slightly younger age compared with pleural mesothelioma and has a slight predilection for women. Few NGS studies of peritoneal mesothelioma have been conducted to date. A first study by Joseph et al. [23], in which peritoneal FFPE specimens from 13 female patients with peritoneal mesothelioma were analyzed using a 510-gene NGS panel (UCSF 500 panel), revealed a high prevalence of *BAP1* mutations (9/13 patients, 69%). Notably, these were mutually exclusive with *NF2* mutations (3/13 patients, 23%). Offin et al. [24] recently published a mono-institutional case series of 50 peritoneal mesotheliomas that underwent genomic sequencing on the Memorial Sloan Kettering Impact Platform (MSK-Impact platform) [25]. This platform sequences all the exons and selected introns in 341 known cancer genes. In this study, mutations occurred most frequently in *BAP1* (30/50 patients, 60%), *NF2* (12/50, 24%), *SETD2* (11/50, 22%) and *TP53* (8/50, 16%), while a low rate of mutations was found in *CDKN2A/B* (<10%), which is different from PM. Tumor mutational burden was confirmed to be low in PeM, with a median number of mutations of 1.8 per megabase. Furthermore, in the peritoneal mesothelioma cohort in the previously cited study [21], the most common alterations detected in peritoneal mesothelioma were inactivating mutations in *BAP1* (47.9%), *NF2* (26.5%), *CDKN2A* (25.9%), *CDKN2B* (19.5%) and *PBRM1* (15.8%). Nevertheless, 1.13% of the patients harbored both rearrangements and short variant alterations in *ALK*, with a potential therapeutic application that will be discussed later (summarized in Table 3).

### 2.3. Pericardial Mesothelioma (PerM)

Pericardial mesothelioma is an exceptionally rare disease; hence, no large population genomics studies have been conducted to date. However, next generation sequencing data from studies by Kato [18], Offin [26] and Schaefer [27] are available. These studies prove that this disease is mainly characterized by mutations in *NF2*, *CDKN2A*, *BAP1* and *TP53*. Notably, a pathogenic germline variant of *BRCA1* (BRCA1 E1210Rfs*), which leads to a biallelic somatic inactivation of *BRCA1* in the tumoral tissue, was found in the study by Schaefer.

### 2.4. Tunica Vaginalis Testis Mesothelioma

Tunica vaginalis testis (TVT) mesothelioma is an ultra-rare cancer occurring mostly in elderly males. Like pericardial mesothelioma, the rarity of the disease reduces the possibility of large population studies. Recently, a mono-institutional study of seven cases of TVT mesothelioma using the targeted NGS panel OncoPanelv3 (Agilent SureSelect, Agilent Technologies, Santa Clara, CA, USA) revealed that the most frequently altered genes were *NF2* (9/13 patients, 71%), *CDKN2A* (6/13 patients, 43%) and *BAP1* (4/13 patients, 29%) [28].

## 3. Targeting Common Mutations in Mesothelioma

### 3.1. CDKN2A

As previously mentioned, *CDNK2A* CNAs leading to gene loss are among the most frequent mutations in mesothelioma occurring at any site. The *CDKN2A* gene encodes two proteins, p16ink4A and p14ARF, with a key role in cell-cycle regulation. The first one acts through the inhibition of cyclin-dependent kinases (CDK) 4 and 6, thus preventing the phosphorylation and activation of the pro-oncogenic retinoblastoma protein (RB). Indeed, p14 inhibits mouse double minute 2 homolog (MDM2), preventing the ubiquitination and degradation of the oncosuppressor p53 [29]. The mesothelioma stratified therapy (MiST) trial is a phase II multi-arm trial that enrolls patients with first-line progressing pleural mesothelioma (NCT03654833). This trial includes a prescreening phase where eligible patients undergo molecular screening of archival tissue using molecular inversion probe-based microarray analysis of somatic copy number aberrations. Then, based on the molecular alterations found, patients are assigned to one of five specific and molecularly matched study arms. Arm 1 of the study involves the treatment of patients with *BAP1* or *BRCA2* alterations using rucaparib; arm 2 involves the treatment of patients with *CDKN2A* alterations using Abemaciclib; arm 3 includes patients with *AXL* alterations who are treated with pembrolizumab plus the AXL inhibitor, bemcetinib; arm 4 includes patients who are treated with atezolizumab plus bevacizumab; and finally, arm 5 includes PDL1-positive *BAP1* deficient or *BRCA2* mutant patients who are treated with niraparib and dostarlimab. The primary endpoint of all the arms in the trial is the disease control rate (DCR) at 12 weeks. In arm 2 of the trial [30], the CDK4/6 inhibitor abemaciclib was tested as a second and subsequent line in PM patients with negative p16ink4A staining based on an IHC assay. A total of 27 eligible patients were screened and 26 were enrolled in the trial. The primary endpoint of the trial of a 12-week DCR was met by 14/26 patients (54%), of whom 3 had PR (11.6%). The median PFS in the trial was 128 days, while the median OS was 217 days. Overall, the treatment was safe, with only one patient experiencing a serious adverse event (diarrhea) from the study treatment (results summarized in Table 4).

### 3.2. BAP1

#### 3.2.1. Targeting BAP1 and DNA Damage Repair Genes (DRR) through PARP Inhibition

Germline mutations leading to the inactivation of *BAP1* (BRCA-associated protein) were associated with familiar pleural mesothelioma almost 15 years ago [39] and subsequent studies revealed a high incidence of *BAP1* inactivation even in sporadic pleural mesothelioma [39,40].

BAP1 protein is a multifunctional tumor suppressor involved in several functions, including chromatin remodeling, DNA damage response through interaction with BRCA1, cell cycle control, cell-death regulation and immune response [41].

BAP1 deficiency may be therapeutically exploited in several ways [41]. Among these, the induction of synthetic lethality through the inhibition of PARP (poli-ADP ribose polymerase) seems to be promising in preclinical models of mesothelioma [42,43] and could thus be exploited in clinics [44]. Furthermore, up to 7% of mesothelioma patients may harbor germline variants in DNA repair complex genes such as *BRCA1*, *BRIP1*, *CHEK2*, *SLX4*, *FLCN* and *BAP1* [45] and these mutations may predispose one to the therapeutic effects of PARP inhibitors.

A few clinical trials have explored the activity of PARP inhibitors in mesothelioma. Olaparib was tested in a mono-institutional phase II trial [31] that enrolled patients with either a peritoneal or pleural mesothelioma progressing to first-line treatment and divided them into three cohorts: (1) germline *BAP1* mutation, (2) somatic *BAP1* mutation or (3) no somatic or germline *BAP1* mutations based on NGS results. The primary endpoint of the trial was ORR. A total of 23 patients were enrolled and treated in this trial. Four of the patients harbored a germline *BAP1* mutation, while eight harbored a somatic *BRCA1* mutation. The primary endpoint of the trial was not met, as only 1 of the 23 patients (4%) showed a partial response, while 18 (78%) had stable disease at 6 weeks and 4 (17%) had progressive disease. The median overall PFS and OS were 3.6 months and 8.7 months, respectively. Interestingly, median PFS and OS in patients with germline *BAP1* mutants were shorter than in patients with the wild-type gene (2.3 versus 4.1 months, *p* = 0.019 for PFS; 4.6 versus 9.6 months, *p* = 0.004 for OS).

A total of 35 patients were screened and 26 were enrolled in arm 1 of the MiST trial, in which eligible patients with either pleural or peritoneal mesothelioma progressing to first-line platinum doublet were treated with 600 mg of rucaparib twice a day. A disease control rate >40% at 12 weeks was the main endpoint of the trial [32]. A total of 15 patients (58%) showed disease stability or a partial response 12 weeks after the start of the trial, with 2/28 patients showing PR. The trial was thus positive based on the prespecified efficacy criteria, while safety was consistent with previous trials of PARP inhibitors in other cancers.

Furthermore, a phase II trial evaluating the PARP inhibitor Niraparib in patients harboring a *BAP1* mutation and a *BAP1* spectrum malignancy showed a disease control rate of 78%, with only 1 PR response [46] (results summarized in Table 4).

Finally, evaluation of PARP inhibitors as therapeutic agents against mesothelioma is ongoing, with the combination of the anti-PDL1 antibody dostarlimab and the PARP inhibitor niraparib currently being tested in arm 4 of the MiST trial and in the UNITO-1 trial. The latter is a phase II trial in which patients with either non-small cell lung cancer or pleural mesothelioma progressing to first-line treatment are screened to find a cohort of patients with over 1% PD-L1 expression and homologous recombination deficiency (HRD). For pleural mesothelioma [47], the study plan is to enroll 35 patients after molecular prescreening. The patients will receive niraparib in association with dostarlimab, with objective response rate as the primary endpoint.

#### 3.2.2. Targeting BAP1 Inactivation through EZH2 Inhibition

In mouse models, the loss of *BAP1* leads to elevated enhancer of zeste 2 polycomb repressive complex 2 subunit (*EZH2*) expression and enhanced repression of polycomb repressive complex 2 (PRC2) targets [48]. The *EZH2* gene codes for a histone methyltransferase that is involved in chromatin remodeling. The inhibition of EZH2 through the administration of the selective inhibitor tazemetostat resulted in enhanced cell death in mesothelioma cell models, either as a single agent [48] or when combined with zoledronate [49]. In a multicentric phase II trial, the EZH2 inhibitor tazemetostat was tested as a single agent in patients with pleural or peritoneal mesothelioma who progressed to first-line treatment [33]. This trial was designed into part 1, where tazemetostat was administered at 800 mg once per day, with pharmacokinetics as a primary objective, and part 2, where tazemetostat was administered at 800 mg twice daily, with disease control rate at 12 weeks as the primary objective. A total of 74 patients were globally enrolled in the trial, including 13 in part 1 and 61 in part 2. From a biomarker point of view, 73/74 (99%) patients had BAP1 protein loss as assessed using immunohistochemistry, and 20/54 (38%) patients had missense or indel inactivating *BAP1* mutations as assessed using somatic DNA sequencing. In part 2 of the trial, 54% (37/61) of the patients had either SD or PR at 12 weeks, with 2 patients showing a partial response. The results were confirmed, considering all the patients enrolled in the trial with a DCR of 51% for all 74 patients, with a 35% median PFS and OS of 18 weeks and 36 weeks, respectively (results summarized in Table 4).

### 3.3. NF2

As previously reported, *NF2*-inactivating mutations are among the most common inactivating mutations arising in pleural mesothelioma. This mutation is also frequent in patients with peritoneal, pericardial and TVT mesothelioma.

*NF2* is a gene that encodes the oncosuppressor protein merlin (Moesin-ezrin-radixin-like protein, also known as schwannomin), a membrane-scaffolding protein that plays a role in several cellular processes by indirectly linking F-actin with transmembrane receptors and intracellular effectors to modulate receptor-mediated signaling pathways such as receptor tyrosine kinase (RTK), cell adhesion, small GTPases, mammalian target of rapamycin (mTOR), focal adhesion kinase (FAK), PI3K/Akt and hippo pathways [50,51].

While direct targeting of the *NF2*-coded protein merlin is difficult, several attempts have been made to target associated downstream signaling pathways such as the PI3K/Akt/mTOR pathway, the focal adhesion kinase (FAK) pathway and the Hippo pathway [52].

#### 3.3.1. PI3K/AKT mTOR

The mTOR (mechanistic Target of Rapamycin) inhibitor everolimus has been tested in pleural mesothelioma patients in the phase II clinical trial SWOG S0722. In this trial, 59 patients failing first-line treatment received 10 mg of everolimus daily until disease progression or intolerable toxicity. The primary objective of the trial was a 6-month PFS rate [34]. The study failed to meet its primary endpoint, with 29% of the patients being alive and progression-free 6 months from the start of the treatment, a disappointing 2% ORR and an OS of 6.3 months. A second study tested the selective dual PI3K/mTOR inhibitor LY3023414 (samotolisib) in 42 patients affected by relapsed pleural mesothelioma [35]. The study was designed as a phase I/II trial whose primary objective in the phase II part was ORR. The study failed to reach its primary endpoint as only 1 patient showed a confirmed partial response, while an additional 2 patients showed an unconfirmed partial response and 17 patients had disease control as a best response. In this trial, molecular NGS profiling was available for 19 patients and 11 harbored a *BAP1* mutation, 5 harbored an *NF2* or *SETD2* molecular aberration; other molecular alterations (mainly *CDKN2A* and *CDKN2B*) were less frequent. The only patient showing a PR had a mutation in *BAP1* (exon 3 p. D34fs) (results summarized in Table 4).

#### 3.3.2. The FAK Pathway

The focal adhesion kinase (FAK), encoded by the *PTK2* gene on chromosome 8, is a non-receptor tyrosine kinase localized to focal adhesions involved in signal conduction from extracellular matrix/integrin engagement [53]. FAK plays a significant role in cell survival, proliferation, motility, migration and invasion. FAK is overexpressed in several malignancies and its overexpression is associated with worse prognosis [54]. A preclinical study performed on mesothelioma cell lines showed that FAK is upregulated and that the selective inhibition of FAK using specific inhibitors leads to a decrease in proliferation as well as increased apoptosis [55]. Furthermore, preclinical evidence supports a specific effect of FAK inhibitors in merlin-deficient PM cell lines and patient-derived xenografts, proving that FAK inhibition leads to significant disease in cell proliferation as well as an increase in apoptosis [56]. Taking into account all the previous evidence, the inhibition of FAK may seem like a logical approach in the treatment of pleural mesothelioma. Nevertheless, evidence of this approach is still underwhelming. In a phase Ib trial, the orally bioavailable FAK inhibitor GSK2256098 was tested in patients with several cancers, mostly mesothelioma (29 patients, 46%) [36]. Despite not being the main endpoint of the trial, limited GSK2256098 activity was demonstrated in mesothelioma patients, with 3 patients showing minor radiologic responses and a median PFS of 12 weeks. Moreover, exploratory translational analysis showed that merlin-negative mesothelioma patients had longer PFS compared with merlin-proficient patients (23.4 vs. 11.4 weeks, respectively). Another phase I trial explored the combination of GSK2256098 and trametinib in a cohort of pretreated patients, of whom 21 (62%) were affected by mesothelioma [37]. No radiological responses were observed, and in mesothelioma, the median PFS was 11.3 weeks. A longer PFS was observed in patients with merlin-negative tumors compared with those with merlin-proficient tumors (15 vs. 7.3 weeks, respectively). Finally, in a phase II trial, defactinib, an orally available high-affinity FAK inhibitor, was tested versus placebo as a maintenance treatment in patients with pleural mesothelioma with at least stable disease after treatment with platinum pemetrexed [38]. The main endpoint of the trial was progression-free survival. The addition of defactinib as a maintenance treatment did not offer an improvement in PFS over placebo (4.1 vs. 4.0 months). Notably, PFS was shorter in merlin-negative patients, both in the defactinib and the placebo groups (4.5 vs. 2.8 months in both arms). Hence, the trial was stopped due to futility and defactinib is not currently being tested in patients with mesothelioma (results summarized in Table 4).

#### 3.3.3. Targeting NF2 Mutations through Hippo Pathway Inhibition

Merlin suppresses the nuclear translocation of YAP (Yes-associated protein 1) and TAZ (tafazzin) through activation of the Hippo pathway. YAP and TAZ are the major effectors of the pathway and associate with TEAD transcription factors in the nucleus to promote the expression of genes involved in cell proliferation and survival [57]. Preclinical experience with a potent TEAD inhibitor, K-975, shows selective cell death in NF2-deficient mesothelioma lines [58]. IAG 933, a potent inhibitor of the interaction between TEAD and the YAP/TAZ complex, is under testing in a phase I trial that is currently enrolling mesothelioma patients or patients with known NF2-inactivating mutations (NCT04857372). Finally, VT3989, a TEAD autopalmitoylation inhibitor, has shown promising preclinical activity in cellular models of mesothelioma [57]. VT3989 was also tested in a Phase I clinical trial where it showed good tolerability and preliminary activity, even in heavily pretreated patients [59].

## 4. Targeting Low Prevalence Alterations in Mesothelioma

### 4.1. ALK Rearrangements

The detection of ALK overexpression in PM using immunohistochemistry was negative in a cohort of 63 patients [60]. Later results from the same group, in which 101 PM patients were analyzed using a FISH-based assay directed at *ALK* or *ROS1*, were consistent and showed that none of the tested tumors harbored an *ALK* fusion [61]. Conversely, *ALK* fusions were found using IHC and FISH screening and subsequent NGS-targeted sequencing of a large retrospective series of 335 PM samples, of which 1 *ALK/EML4* fusion [62] was reported in at least two patients [63,64]. Notably, in one of these reports, the activity of the ALK inhibitors, alectinib and ceritinib, was demonstrated [64].

A retrospective study of 88 patients with peritoneal mesothelioma demonstrated a low prevalence of *ALK* fusions (3/88, 3.4%), including 1 in a novel and previously unreported fusion partner (*ATG16L1*) [65]. Notably, all patients with *ALK* fusions were women, were not exposed to asbestos and were significantly younger than other patients (32 vs. 69 years). Crizotinib showed clinical activity in some of the *ALK*-rearranged peritoneal mesotheliomas [66]. Finally, a growing body of evidence demonstrates that pediatric patients affected by peritoneal mesothelioma may have *ALK*-rearranged tumors that may have prolonged clinical benefit from the administration of crizotinib or ceritinib, lasting up to 10 years in the case of a girl treated with crizotinib [67,68].

### 4.2. KRAS Mutations

Selective KRAS inhibitors have recently proved efficacious in treating tumors where a *KRAS* mutation (G12C in most cases) was detected [69]. As previously reported, 1–2% of patients with PM or PeM may have a potentially actionable KRAS mutation and thus could be treated with specific KRAS inhibitors. Furthermore, high throughput screening methods and preclinical models indicate KRAS as a potentially actionable mutation, at least in PM patients [70,71].

### 4.3. Hedgehog Pathway Mutations

Preclinical evidence suggests an overexpression and a direct involvement of the hedgehog (HH) signaling pathway in the proliferation of pleural mesothelioma cell lines [72,73]. In addition, a potential role of SMO inhibitors, such as vismodegib, in inhibiting this cellular proliferation in murine models of PM has been previously demonstrated [74].

Recently, a case report of a patient with pleural mesothelioma with a *PTCH1* F1147fs mutation indicated sustained and long-lasting activity of the SMO inhibitor vismodegib [75]. *PTCH1* is a gene that encodes an effector of the HH pathway whose overexpression is associated with uncontrolled proliferation in pleural mesothelioma cell lines. The administration of the SMO inhibitor vismodegib was associated with a decrease in tumor growth and a reduction in tumor progression in a rat model of mesothelioma [74].

### 4.4. PDGFRA and PDGFRB

As previously stated, up to 2% of pleural mesothelioma patients may have pathogenic and potentially actionable mutations in PDGFR receptors. From a preclinical point of view, mesothelioma cell lines may overexpress *PDGFRB,* and the combination of gemcitabine and imatinib, a multikinase inhibitor targeting KIT, BCR-ABL fusion protein and PDGFRB, induced cell death and reduced tumor growth in cellular lines [76] and xenografted murine models of mesothelioma overexpressing *PDFGRB* [77]. Despite the promising activity demonstrated by in vitro and in vivo models, imatinib did not show significant activity when tested in clinical trials, both as a single agent and in combination with gemcitabine. Indeed, the ORR in phase II trials using imatinib as a single agent in relapsed mesothelioma was 0% [78,79], while a phase II trial assessing the combination of gemcitabine and imatinib in pretreated PM patients failed to meet the prespecified primary endpoint of the study (PFS > 75% at 3 months); the trial was therefore deemed formally negative.

### 4.5. FGFR

In a drug screening assay, a mesothelioma cell line harboring a *BAP1* mutation that did not have mutations in any of the *FGFR* genes but showed functional overexpression of the FGFR pathway was shown to be sensitive, both in vitro and in vivo, to the inhibition of the FGFR pathway mediated by the irreversible FGFR inhibitor AZ4547 [80]. Despite this clinical evidence and the fact that up to 1% of mesothelioma patients might harbor potentially actionable *FGFR* mutations, the only patient with mesothelioma enrolled in RAGNAR, a phase II trial of erdafitinib in solid tumor with targetable FGFR1/4 alterations (mutations and fusions), did not show a tumoral response when they were treated with the pan-FGFR inhibitor erdafitinib [81].

## 5. Conclusions

In this review, we summarized the available evidence on the clinical application of NGS in mesothelioma and its potential relevance in identifying actionable therapeutic targets for this disease. We found that NGS studies performed across all mesothelioma subtypes are concordant in demonstrating a disease that is mainly characterized by inactivating mutations in oncosuppressor genes.

Currently, the only way of inhibiting these aberrant proliferative pathways is downstream targeting of deregulated proliferative factors. Although this seems like a logical way to treat mesothelioma, most phase II studies using this strategy have failed to meet their primary endpoints.

The reason for this lack of efficacy may lie in inappropriate biomarker stratification (i.e., the use of immunohistochemistry instead of NGS methods for patient screening) or an intrinsic lack of affinity for targets by the investigated drugs. Furthermore, a growing body of evidence points to mesothelioma as a highly genomically heterogeneous disease that may partially explain the limited effectiveness of drugs directed toward a specific target [82].

In addition, objective response rate might not be the best endpoint for defining the efficacy of systemic agents in the treatment of mesothelioma, especially in the refractory disease setting. This might be due to the objective difficulty of measuring mesothelioma since this disease often presents as a diffuse thickening of a serous cavity rather than a mass-forming disease. Clinical benefit may conversely be a better outcome for the development of drugs against mesothelioma, as this is a symptomatic disease where symptom reduction can play a significant role.

Conversely, a low prevalence (1–2%) of activating mutations in oncogenes may occur in some patients. Evidence of the therapeutic exploitation of these mutations is scarce and consists mainly of case reports. Nevertheless, taking into account the rarity of mesothelioma in general and thus the ultra-rarity of these low prevalence mutations, the same principles for the treatment of ultra-rare sarcomas [83] might be translated in clinical practice for mesothelioma patients harboring very rare mutations.

Machine learning and artificial intelligence might further help in the development of novel predictive models for finding these rare mutations by identifying patients that are more likely to harbor targetable mutations, similar to what has already been highlighted for prognostic models of pleural mesothelioma based on gene mutations [84].

## 6. Future Perspectives

With the exceptions of clinical trials involving anti-VEGF/VEGFR2 agents, most mesothelioma trials involving targeted agents have shown disappointing efficacies. Nevertheless, most of the published arms of the MIST trial, where immunohistochemistry was used to enroll patients into specific trial arms, proved that molecular selection may help in identifying patients for whom a targeted approach might be feasible. We advise that future clinical trials of mesothelioma involving targeted agents should be based on refined molecular stratification based on scales of target actionability such as the ESCAT scale rather than on a “one size fits all” approach.

On the other hand, the finding that 1–2% of mesothelioma patients may bear potentially actionable molecular alterations such as ALK rearrangements is relevant for clinical practice, as some patients might benefit from direct targeting of such alterations. NGS may thus be able to find patients for whom such treatment is appropriate. Nevertheless, NGS is expensive and not routinely affordable at every institution, particularly taking into account the rising incidence of mesothelioma in developing countries. In our opinion, to maximize the impact of NGS in identifying patients with actionable molecular targets in mesothelioma, identification of clinical variables associated with higher probabilities of finding such alterations is mandatory.

Another application of clinical NGS in patients with mesothelioma is the identification of potentially actionable immune checkpoints. As an example, the overexpression of the V-domain Ig suppressor of T-cell activation, VISTA, has been observed in pleural mesothelioma. VISTA is a potentially actionable immune checkpoint and to date, two drugs, the monoclonal antibody JNJ-61610588 and the orally available small molecule inhibitor C170, have been tested against mesothelioma [85].

Finally, we think that large international collaborations within reference networks such as Euracan (European Rare Cancer Networks) can serve as a basis for the development of international collaborations aimed at improving these therapeutic and diagnostic pitfalls in mesothelioma.

## Figures and Tables

**Table 1 cancers-15-05716-t001:** Whole exome sequencing of mesothelioma.

References	No of Patients	Most common genomic alterations
Guo et al. [13]	22	Protein modifications: *BAP1*, *NF2* and *CUL1* *CNAs: CDKN2A*, *NF2* and *BAP1*
Bueno et al. [14]	211 total 99 WES	*BAP1*, *NF2*, *TP53*, *SETD2*, *DDX3X*, *ULK2*, *RYR2*, *CFAP45*, *SETDB1* and *DDX51*
Hmeljak et al. [15]	74	*BAP1*, *NF2*, *TP53*, *LATS2* and *SETD2*

**Table 2 cancers-15-05716-t002:** Molecular alterations identified in PM using targeted NGS.

Study	No of Patients	Panel Used	Relevant Therapeutic Alterations
Shukuya et al. [16]	42	Targeted amplicon-based cancer hotspot panel	3 mutations in PDGFRA 1 mutation in KRAS
Lo Iacono et al. [17]	123 all stage > III	Custom NGS 50-gene NGS panel + NF2 and BAP1	None
Kato et al. [18]	23	FoundationOne [19]	All patients had at least one theoretically targetable alteration
Hiltbrunner et al. [21]	1113	FoundationOne [22]	<1%: KRAS, EGFR, PDGFRA/B, ERBB2 and FGFR3 1–2%: ALK, PTCH1, SUFU and BRCA2.

**Table 3 cancers-15-05716-t003:** Molecular alterations in PeM identified using targeted NGS.

Study	No of Patients	Panel Used	Relevant Therapeutic Alterations
Joseph et al. [23]	13	510-gene targeted NGS panel (UCSF panel)	Not reported
Offin et al. [24]	50	MSK Impact	Not reported
Hiltbrunner et al. [21]	355	FoundationOne [22]	1–2%: ALK

**Table 4 cancers-15-05716-t004:** Summary of the main alterations in mesothelioma and relative targeted agents trial.

Gene	Type of Molecular Alteration	Alteration Prevalence (%)	Drug Tested, Primary Endpoint, Phase Trial
*CDKN2A*	Deletions, allelic losses, somatic inactivations	Pleural: 45–49% [13,14,15,16,17,18,20,21] Peritoneal: <10–26% [21,23,24]	– Abemaciclib, DCR, 12-week DCR (54%), positive trial [30]
*BAP1*	Somatic-inactivating mutations	Pleural: 23–57% [13,14,15,16,17,18,20,21] Peritoneal: 48–60% [21,23,24]	PARP – Olaparib, phase II ORR (4%), negative trial [31] – Rucaparib, phase II, DCR (54%), positive trial [32] – Tazemetostat, phase II, DCR (54%), positive trial [33]
*NF2*	Somatic-inactivating mutations, deletions, allelic losses	Pleural: 23–30% [13,14,15,16,17,18,20,21] Peritoneal: 27% [21,23,24]	PI3K/AKT/mTOR – Everolimus, phase II, 6 months PFS (29%), negative trial [34] – Samotolisib, Phase I/II, ORR (2%), negative trial [35] – FAK – GSK2256098, Phase I, negative [36] – GSK2256098 plus trametinib, Phase I, negative [37] – defactinib, phase II, no PFS improvement over placebo, negative [38]

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
