# Peer review of "Clinical Next Generation Sequencing Application in Mesothelioma: Finding a Golden Needle in the Haystack"

_cancers, 2023, doi:10.3390/cancers15245716_

Round 1

Reviewer 1 Report

Comments and Suggestions for Authors

This review has no novel progress in mesothelioma.

Author Response

We would like to have a discussion via email with the reviewer about the reason for his/her firm rejection of our paper.

Reviewer 2 Report

Comments and Suggestions for Authors

Dr. Cerbone and colleagues discuss the use of clinical next generation sequencing for mesothelioma. I have the following specific comments:

1)      In the introduction, why is non-small cell lung cancer and melanoma capitalized but cholangiocarcinoma is not?

2)      Missing key reference from MSK PMID 34332931.

3)      Should include reference to Vivace trial activity of TEAD inhibitor which was presented at AACR 2023 Annual Meeting and WCLC IASLC 2023.

4)      I think the discussion of why targeted therapy trials have had underwhelming results to do date could be further expanded.

Comments on the Quality of English Language

The manuscript is well written and clear.

Author Response

Dear Reviewer, 

On the behalf of all the coauthors of the paper I would first of all thank you for your interesting commentaries. We added to the paper all the paper you suggested and also expanded the discussion including.

Many thanks  

LC

Reviewer 3 Report

Comments and Suggestions for Authors

It’s an interesting review, regarding target therapy and genomic study of mesothelioma. Mesothelioma featured with tumour suppressor gene mutations, which leads to difficulties in target therapy development. The authors summarized the gene alterations of mesotheliomas by NGS and related target therapy development for different types of mesotheliomas.

I have some major comments as follows:

1.       Some of reference citations were shown improperly. Such as line 111, ‘reported19’ should be ‘reported19’, and line 196, ‘ways34’. Please check the format of citations over the paper.

2.       It will be great if the authors provided a summary table to summarize all the mutations of mesothelioma, corresponding therapy development (which drug, at which phase, for which subtype of mesotheliomas) and references, described in the review.

3.       Line 246, the authors mentioned ‘tazemetostat was administered at 800 mg twice daily’. The dosage is 800 mg? I think it should be 800 mg/m2. Also line 218 (600 mg), and line 271 (10 mg)

4.       The authors mentioned VISTA is abundantly expressed in mesothelioma in section 2.1.1. Is there treatment available in research related to this?

5.       Be aware of gene or protein name spelling. ‘Axl alteration’ should be ‘AXL alteration’. ‘axl’ inhibitor should be ‘AXL inhibitor’, in line 176. Please check the spelling and format over the manuscript.

Comments on the Quality of English Language

English is good. Check the spelling mistakes. Should it be capital or lowercase? Gene names should be Italic. Protein names should be all capital for human.

Author Response

Dear Reviewer, 

I would like to thank you on the behalf of all the coauthors for all of your interesting remarks.

Concerning remark 1 and 5 we corrected all the typos you have identified

concerning remark 2, we added table 4 in which the results of the main trial based on target identification in mesothelioma are summarized

concerning remark 3, we would to thank you for your commentary, nevertheless as VISTA is an immune  checkpoint, so to the best of our knowledge, the addition of a paragraph for VISTA might be out of the main scope of this review. Hence we added a small reference to VISTA in the conclusion of the paper.

Concerning remark 4, we carefully checked to the best of our knowledge the dosage of the drugs listed (tazemetostat, niraparib and everolimus) is the one we cited in the paper. 

Thanks again for your remarks

LC